# Unsupervised Hierarchical Concept Learning

## Abstract

Concepts or temporal abstractions are an essential aspect of learning among humans. They allow for succinct representations of the experiences we have through a variety of sensory inputs. Also, these concepts are arranged hierarchically, allowing for an efficient representation of complex long-horizon experiences. Analogously, here we propose a model that learns temporal representations from long-horizon visual demonstration data and associated textual descriptions without explicit temporal supervision. Additionally, our method produces a hierarchy of concepts that align more closely with ground-truth human-annotated events than several state-of-the-art supervised and unsupervised baselines in complex visual domains such as chess and cooking demonstrations. We illustrate the utility of the abstracted concepts in downstream tasks, such as captioning and reasoning. Finally, we perform several ablation studies illustrating the robustness of our approach to data-scarcity.

## 1 Introduction

Consider a video (Figure 1) that demonstrates how to cook an egg. Humans subconsciously learn concepts (such as boiling water) that describe different concepts (or skills) in such demonstrations Pammi et al. (2004). These learned skills can be composed and reused in different ways to learn new concepts. Discovering such concepts automatically from demonstration data is a non-trivial problem. Shankar et al. (2019) introduces a sequence-to-sequence architecture that clusters long-horizon action trajectories into shorter temporal skills. However, their approach treats skills as independent concepts. In contrast, humans organize these concepts in hierarchies where lower-level concepts can be grouped to define higher-level concepts Naim et al. (2019).

We extend the architecture in Shankar et al. (2019) to simultaneously discover concepts along with their hierarchical organization without any supervision.

We propose an end-to-end trainable architecture UNHCLE for hierarchical representation learning from demonstrations. UNHCLE takes as input a long horizon trajectory of high-dimensional images demonstrating a complex task (in our case, chess and cooking) and the associated textual commentary and isolates semantically meaningful subsequences in input trajectories. We emphasize that it does not require temporal annotations which link subsequences in the trajectories of images to the free-flowing commentary, but instead, autonomously discovers this mapping. Therefore, this work takes a step towards unsupervised video understanding of high-dimensional data. Our contributions can be summarized as follows:

- We introduce a transformer-based architecture to learn a multi-modal hierarchical latent embedding space to encode the various concepts in long-horizon demonstration trajectories. UNHCLE abstracts these concepts (shown through visual qualitative analysis) without requiring any temporal supervision, i.e., it divides long-horizon trajectories into semantically meaningful subsequences, without access to any temporal annotations that split these trajectories optimally.

- We show the quantitative effectiveness of learning high-level concepts in a hierarchical manner compared to learning them in isolation while outperforming several baselines on YouCook2 (Zhou et al. (2017)) and Chess Opening dataset[1].

- We further introduce a mechanism to incorporate commentary accompanying demonstrations in UNHCLE and show improvements in hierarchical concepts discovered.

---

[1]https://www.kaggle.com/residentmario/recommending-chess-openings

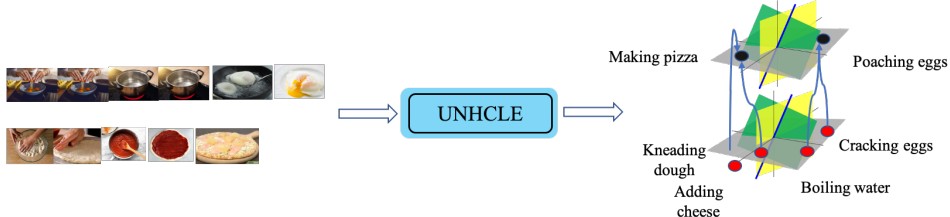

**Figure 1:** Overview of our approach. UNHCLE learns a hierarchical latent space of concepts describing long horizon tasks like cooking and chess gameplay.

- We introduce TimeWarped IoU (TW-IoU), an evaluation metric that we use to compare the alignment of our discovered concepts and ground-truth events.

Existing approaches to representation learning for demonstrations or videos typically require significant supervision. Typically, sequence-to-sequence architectures are trained on datasets segmented by humans. During inference, these architectures generate proposals for timestamps that segment the input trajectory into semantically meaningful sequences. These complex sequence-to-sequence models require significant amounts of annotated data, making them costly to train them.

More generally, video and scene understanding is an important research area with wide-ranging applications. Most recently, Chen et al. (2019) utilize semantic awareness to perform complex depth estimation tasks to acquire the geometric properties of 3-dimensional space from 2-dimensional images. Tosi et al. (2020) utilize similar semantic information for depth estimation, optical flow and motion segmentation. Boggust et al. (2019) attempt to ground words in the video, but apply significant supervision to synchronize them, requiring human intervention. We attempt to learn similar embeddings but do so in a completely unsupervised manner, not utilizing any of the temporal labels available.

The field of learning from demonstrations (Nicolescu & Mataric (2003)) seeks to learn to perform tasks from a set of demonstrated behaviors. Behavioral Cloning is one popular scheme (Esmaili et al. (1995)). Atkeson & Schaal (1997) and Schaal (1997) show how agents can learn simple tasks like cartpole simply from demonstrations. Pastor et al. (2009) also study how robots can learn from human demonstrations of tasks. Peters et al. (2013) and Kober & Peters (2009) fit a parametric model to the demonstrations. Niekum et al. (2012), Murali et al. (2016), and Meier et al. (2011) first segment trajectories into subsequence and then apply a parametric models to each subsequence. More recently, Schmeckpeper et al. (2019) shows that agents can learn to maximize external reward using a large corpus of observation data, i.e., trajectories of states on a relatively smaller corpus of interaction data, i.e., trajectories of state-action pairs. Hierarchical task representations have been studied as well. Instead of treating demonstrations in a flat manner, one may also infer the hierarchical structure. A few recent works attempt to do so (Xu et al. (2018); Sun et al. (2018)), or as task graphs (Huang et al., 2019). Both Xu et al. (2018) and Huang et al. (2019) address generalizing to new instances of manipulation tasks in the few-shot regime by abstracting away low-level controls. However, all of these approaches require an environment i.e., a transition and reward function to learn from. On the contrary, humans show an ability to learn by watching demonstrations, which we attempt to replicate.

Temporal abstractions of action sequences or skill/primitive learning is also a related field. Eysenbach et al. (2018), learn a large number of low-level sequences of actions by forcing the agent to produce skills that are different from those previously acquired. However, due to the diversity bias, the agent results in learning many useless skills that cannot be used for any semantically meaningful task. Similarly, Sharma et al. (2019) attempt to learn skills such that their transitions are almost deterministic in a given environment. These approaches also require access to an environment, whereas we try to learn without an environment.

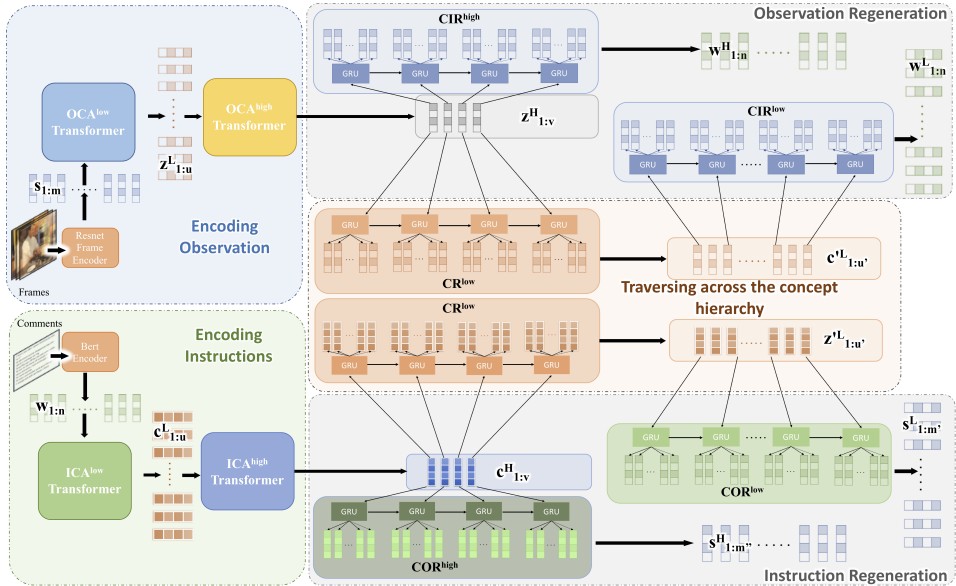

**Figure 2:** Overview of our approach. UNHCLE learns a hierarchical semantically-meaningful embedding space which allows it to perform complex downstream tasks such as temporal concept segmentation and label prediction. Refer to Section 2.1 for details about the abbreviations used in the figure.

## 2 APPROACH

### 2.1 UNHCLE: UNSUPERVISED HIERARCHICAL CONCEPT LEARNING

Intuitively, we define a concept as a short sequence of states which repeatedly occur across several demonstration trajectories. Concepts have an upper limit on their length in time-steps. These concepts can be obtained from the images of demonstrations, denoted by $z$, and from the associated textual description, represented by $c$. We also refer to them as Observational Concept (OC) and Instructional Concept (IC) respectively and the module that encodes these concepts is referred to as Observational Concept Abstraction (OCA) and Instructional Concept Abstraction (ICA) modules. Additionally, concepts are hierarchical in nature - thus, lower-level and higher-level observational or image concepts are denoted by $z^L$ and $z^H$, respectively. Analogously, lower-level, and higher-level textual concepts are represented by $c^L$ and $c^H$, respectively. Similarly, the higher-level and lower-level modules are denoted by $low$ or $high$ for the corresponding level.

Once we obtain these concept abstractions across levels, we can also transform them across the levels using a Concept Regeneration (CR) Module to transform $low$-level concepts to $high$-level concepts and vice-versa. Instead of just traversing in the concept level hierarchy, we can also use the Concept Instruction Regeneration Module or CIR to obtain the original instructions that map to that concept in its respective concept modality.

Subsequently, we provide details about the different stages in our proposed technique.

**Encoding Observation:** Given a long horizon trajectory of demonstration images along with its associated textual description, UNHCLE is able to abstract a hierarchy of concepts from demonstration images. We first pass these input images through ResNet-32 (He et al. (2016)) to get a sequence of image vectors as $S = s_{1:m}$, and the associated text is converted into word vectors $W = w_{1:n}$ using BERT-base (Devlin et al. (2018)). Observations combine to produce lower-level concepts whereas higher-level concepts are simply aggregations of such lower-level concepts. Thus, the Lower-level Observation Concept Abstraction module (OCA$^{low}$) is trained to embed a sequence of image vectors of a video $(s_1, s_2....s_m)$ into a sequence of concept vectors $(z_1^L, z_2^L, .., z_U^L)$ where $u << m$ such that $z_{1:u}^L = OCA^{low}(s_{1:m})$. Subsequently, lower-level concepts combine together to form a higher level of concepts using the Higher-level Observation Concept Abstraction module (OCA$^{high}$) such that $z_{1:v}^H = OCA^{high}(z_{1:u}^L)$.

**Encoding Instructions:** We also endeavour to discover these higher-level and lower-level concepts through natural language instructions. The Lower-level Instruction Concept Abstraction module ($\text{ICA}^{low}$) and Higher-level Instruction Concept Abstraction module ($\text{ICA}^{high}$) are responsible for this functionality. From a corpus of words, $(w_1, w_2....w_n)$, the $\text{ICA}^{low}$ module generates concept $(c_1^L, c_2^L, .., c_u^L)$, $u << n$: $c_{1:u}^L = \text{ICA}^{low}(w_{1:n})$. Subsequently the $\text{ICA}^{high}$ encodes the lower-level language concepts $c_{1:u}^L$ into higher level concepts as $c_{1:v}^H = \text{ICA}^{high}(c_{1:u}^L)$.

**Traversing across the concept hierarchy:** Learning concepts at different levels has an added advantage of traversing these concepts in the concept hierarchy. This additionally allows us to utilize these hierarchical traversals to obtain coarse or fine-grained concepts at any level. We can thus regenerate the lower concepts from higher-level instruction concepts using the Lower-level Concept Regeneration Module ($\text{CR}^{low}$) such that $z_{1:u'}'^L = \text{CR}^{low}(c_{1:v}^H)$. We can then later utilize this to obtain lower-level concepts from the higher-level concepts to regenerate the demonstration images $S^L = s_{1:m^L}^L$ in a cross-modal fashion.

**Observation and Instruction Regeneration:** Under a concept, the sequence of frames is nearly deterministic i.e. the knowledge of a concept uniquely identifies the accompanying sequence of images in the demonstration trajectory. Subsequently, we regenerate the demonstration image vectors $S^L = s_{1:m^L}^L$ from lower-level concepts using Lower-level Concept Observation Regeneration Module ($\text{COR}^{low}$) such that $s_{1:m^L}^L = \text{COR}^{low}(z_{1:u'}'^L)$. We also regenerate the demonstration image vectors $S^U = s_{1:m^U}^U$ from higher-level concepts abstracted from language using Higher-level Concept Observation Regeneration Module ($\text{COR}^{high}$) such that $s_{1:m^U}^U = \text{COR}^{high}(c_{1:v}^H)$ in a similar cross-modal manner.

Finally, inspired by humans who can easily describe concept representations using free-flowing natural language, we first regenerate lower-level concepts from higher-level observation concepts using the Lower-level Concept Regeneration Module ($\text{CR}^{low}$) such that $c_{1:u'}'^L = \text{CR}^{low}(z_{1:v}^H)$, and subsequently, regenerate the word vectors $W^L = w_{1:n^L}^L$ from lower-level concepts using Lower-level Concept Instruction Regeneration Module ($\text{CIR}^{low}$) such that $w_{1:n^L}^L = \text{CIR}^{low}(c_{1:u'}'^L)$. Additionally, the higher-level concepts identified by the $\text{OCA}^{high}$ module from demonstration frames are also described using a meaningful free-flowing commentary by the Higher-level Concept Instruction Regeneration module or $\text{CIR}^{high}$. Thus, we regenerate the word vectors $W^U = w_{1:n^U}^U$ from higher-level concepts using $\text{CIR}^{high}$ such that $w_{1:m^U}^U = \text{CIR}^{high}(z_{1:v}^H)$.

## 2.2 SOFT-DYNAMIC TIME WARPING (SOFT-DTW)

The most significant challenge that our work overcomes is that we do not require supervision for hierarchical temporal segmentation i.e. we do not require annotations which demarcate the beginning and ending of a concept, both in language and in the space of frame's timestamps. Currently, most approaches rely on annotation describing the frame start and end for an event and the associated text for this event. Our architecture is trained using free-flowing commentary without event demarcations. This is achieved using the loss function we implement. We utilize Soft-DTW (Cuturi & Blondel (2017)) loss to align the reconstructed trajectories with the input long-horizon trajectory. So given two trajectories $x = (x_1, x_2, ...x_n)$ and $y = (y_1, y_2, ...y_m)$, the soft-DTW$(x, y)$ is computed as,

$$\text{soft-DTW}(x, y) = min^\gamma \{\langle \mathcal{A}, \ \Delta(x, y)\rangle, \ \mathcal{A}\} \tag{1}$$

where $\mathcal{A} \in \mathcal{A}_{n,m}$ is the alignment matrix, $\Delta(x, y) = [\delta(x_i, y_i)]_{ij} \in \mathbb{R}^{n \times m}$ and $\delta$ being the cost function. $min^\gamma$ operator is then computed as,

$$min^\gamma \{a_1, \cdots, a_n\} = \begin{cases} min_{\,i \leq n} \ a_i, & \gamma = 0, \\ -\gamma \log \sum_{i=1}^n e^{-a_i/\gamma}, & \gamma > 0. \end{cases} \tag{2}$$

For our experiments, we use $L_2$ distance as $\delta$ and $\gamma = 1$.

## 2.3 LEARNING OBJECTIVE

Our approach that has been outlined in Figure 2 uses several loss terms between different network outputs to achieve our objective. The soft-DTW is used between several sequences as follows.

$$\mathcal{L}_{dyn} = \text{soft-DTW}(z_{1:u}^L, c_{1:u'}'^L) + \text{soft-DTW}(c_{1:u}^L, z_{1:u'}'^L) + \text{soft-DTW}(s_{1:m}, s_{1:m'}'^L) + \text{soft-DTW}(s_{1:m}, s_{1:m''}'^H)$$

We use the Negative Log-Likelihood (NLL) loss ($\mathcal{L}_{nll}$) between the generated comment vectors and the BERT vectors and use the $L_2$ loss between concepts from different modalities as follows,

$$\mathcal{L}_{static} = \mathcal{L}_{nll}(w_{1:n}, w'^L_{1:n}) + \mathcal{L}_{nll}(w_{1:n}, w'^H_{1:n})$$

We then define our final loss as, $\mathcal{L}_{total} = \alpha * \mathcal{L}_{dyn} + \beta * \mathcal{L}_{static}$

## 2.4 EVALUATION METRICS

The ground-truth events in the dataset and the concepts generated by UNHCLE may differ in number, duration, and start-time. To evaluate the efficacy of UNHCLE in generating concepts that align with the human-annotated events in our dataset, it is imperative that we utilize a metric that measures the overlap between generated concepts and ground truths and also accounts for this possible temporal mismatch. To this end, we introduce time-warp IoU. Note that we do not utilize these the temporal ground-truth event annotations during training, but only to compare the abstraction generated by our architecture to the human-annotated events.

Consider the search series $S = (s_1, s_2, s_3 ... s_M) \in \mathbb{S}$ and target series $T = (t_1, t_2, t_3 ... t_N)$ where $S$ corresponds to the end-of-concept time stamp for each concept as generated by UNHCLE for a single long-horizon demonstration trajectory. Thus, the $i^{th}$ concept abstracted from UNHCLE starts at time = $s_{i-1}$ and end at time = $s_i$. Similarly, $T$ corresponds to the end-of-event time stamp for each ground-truth event in the demonstration trajectory, where the $j^{th}$ ground truth event starts at time = $t_{j-1}$ and ends at time = $t_j$. Note that both $s_0$ and $t_0$ are equal to zero i.e. we measure time starting at zero for all demonstration trajectories.

### 2.4.1 TIME-WARPED IOU

We calculate the time-warped alignment between between the event annotations and the concepts from UNHCLE. This implies calculating $\Delta(S, T)$, solving the following optimization problem (Berndt & Clifford (1994)), $\Delta(S, T) = \min_{P \in \mathcal{P}} \sum_{m,n \in P} \delta(s_m, t_n)$, where the $S$ and $T$ correspond to the search and target series respectively and $\delta$ corresponds to a distance metric (in our case the $L_2$ norm), measuring time mismatch.

$\Delta(S, T)$ therefore corresponds to the trajectory discrepancy measure defined as the matching cost for the optimal matching path $P$ among all possible valid matching paths $\mathcal{P}$ (i.e. paths satisfying monotonicity, continuity, and boundary conditions). From this optimal trajectory obtained we can also get the warping function $W$ such that $W(s_i) = t_j$, i.e. we find the optimal mapping between the $i^{th}$ concept ending at time = $s_i$ and the $j^{th}$ event ending at time = $t_j$. To quantify the quality of this mapping we introduce Time-Warp Intersection over Union for a single long-horizon trajectory $S$.

$$TW - IoU(S) = \sum_{t_i} \frac{\sum_{s_j : W(s_j) = t_i} \min(t_i, s_j) - \max(t_{i-1}, s_{j-1})}{\max_{s_j : W(s_j) = t_i} \{\max(t_i, s_j)\} - \min_{s_j : W(s_j) = t_i} \{\min(t_{i-1}, s_{j-1})\}} \quad (3)$$

Intuitively, this corresponds to the overlap between the $i^{th}$ concept ending at time = $s_i$ and the $j^{th}$ event ending at time = $t_j$.

## 3 EXPERIMENTS

### 3.1 DATASETS

In order to find conceptual temporal patterns from videos in a hierarchical manner, we experiment with two datasets from different domains.
**YouCook2** (Zhou et al. (2017)) dataset comprises of instructional videos for 89 unique recipes ($\sim$22 videos per recipe) containing labels that separate the long horizon trajectories of demonstrations into events - with explicit time stamps for the beginning and end of each event along with the associated commentary. It contains 1,333 videos for training and 457 videos for testing. The average number of segments per video is 7.7 and the average duration of the video is 5.27 minutes.
**Recommending Chess Openings**[2] dataset consists of opening moves in the game of Chess. An

---

[2]https://www.kaggle.com/residentmario/recommending-chess-openings

Opening in Chess is a fixed sequence of moves which when performed leads to a final board state putting the player in a strategic position in the game. Commonly used chess openings are each labeled with a name (Figure 3 shows examples). The dataset contains 20,058 openings with each containing a sequence of chess moves and it's corresponding opening and variation labels. The train-test split used for our experiments is 80-20.

## 3.2 IMPLEMENTATION DETAILS

The Observational Concept Abstraction (OCA) and Instructional Concept Abstraction (ICA) modules consist of the Transformer (Vaswani et al. (2017)) Encoder with 8 hidden layers and 8-Head Attention which takes as input, a sequence of frames positionally encoded and outputs a hidden vector. It is then passed through a Transformer Decoder again having 8 hidden layers and encoder-decoder cross-attention with masking which finally outputs the concepts which are latent variables having dimension $conceptlength \times 768$. We use a 1-layer GRU (Chung et al. (2014)) for our Concept Instruction Regeneration (CIR), Concept Observation Regeneration (COR), and the Concept Regeneration (CR) module which converts the high-level concepts to low-level concepts. For encoding the image frames, we down-sample all videos to 500 frames and use ResNet-32 (pretrained on MSCOCO dataset) (He et al. (2016)) for dense embedding of dimension $512 \times 1$. We further down-sample to 200 frames per trajectory for all experiments. Comments are encoded using BERT-base pre-trained embeddings with a 768 hidden dimension.

For our low-level concepts, we keep the maximum number of concepts discovered as 16 and for high-level concepts as 4. These assumptions are based on the YouCook2 dataset statistics wherein ground-truth annotations (not used during training) the minimum number of segments were 5 and the maximum as 16. We keep $\alpha = 1$ and $\beta = 1$ for all our experiments along with a batch-size of 128 and use Adam optimizer for training with a learning rate of $1e - 5$ and train the network for 100 epochs.

## 3.3 RESULTS AND ANALYSIS

### 3.3.1 VISUALIZING HIERARCHY

Here we analyze whether the discovered concepts are human interpretable i.e. *are the temporal clusters within a single demonstration semantically meaningful?*. We find that our architecture abstracts several useful human interpretable concepts without any supervision. Figure 3 and 4 shows the results obtained. These abstracted high-level concepts align well with the ground truth event labels provided in YouCook2. We also additionally find that our model is able to split each ground truth event into lower-level concepts. For instance, in a pasta-making demonstration in YouCook2, a single event corresponding to the description "*heat a pan add 1 spoon oil and prosciutto to it*", UNHCLE is divided into low level concepts corresponding to "*heat pan*", "*add oil*" and "*prosciutto*". Note that no explicit event time labels were provided to UNHCLE, indicating that our model is able to abstract such coherent sub-sequences, thus taking the first step towards video understanding.

We also visualize the concepts abstracted by UNHCLE when trained on chess opening data. The concepts learnt here also produce coherent, human-interpretable results.

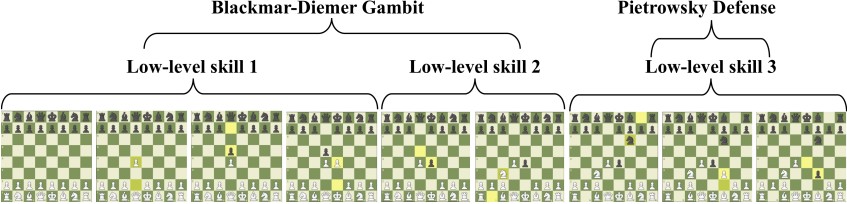

**Figure 3:** Hierarchy of concepts discovered by UNHCLE using openings data in Chess. At a high level, UNHCLE correctly identifies concepts corresponding to the Blackmar-Diemer Gambit and the Pietrowsky Defense. At a low level, it identifies concepts such as "*d4 d5..* and *e4 d3 ..*" that are used across several openings

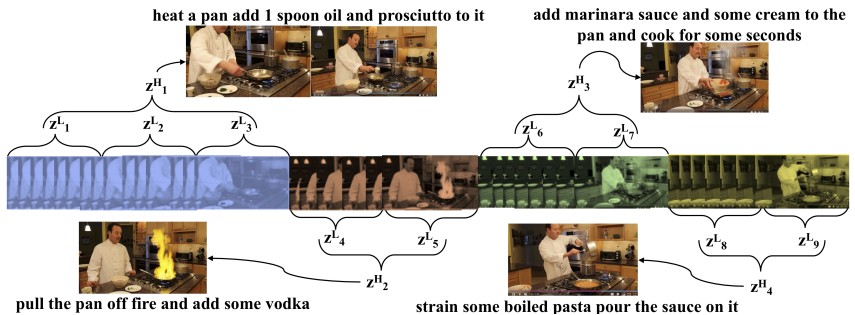

**Figure 4:** Example Hierarchy of concepts discovered by UNHCLE on the YouCook2 dataset

### 3.3.2 COMPARISON WITH BASELINES

In this section, we evaluate the performance of UNHCLE quantitatively on YouCook2[3] and quantify its ability to generate coherent concepts that align with the human annotated ground truths using the TW-IoU metric. We compare our approach with 4 baselines. (1) **Random** baseline predicts segment randomly on the basis of uniformly sampled timestamps for a given video on the basis of its duration. (2) **EQUALDIV** consists of dividing the video into conceptual segments of equal duration (3) **GRU-supervised**: Further we also consider a naive supervised baseline comprising of a GRU Cho et al. (2014) based encoder that sequentially processes the Resnet features corresponding to frames in a video followed by a decoder GRU that attends Bahdanau et al. (2014) on encoder outputs and is trained to sequentially predict end timestamp of each meaningful segment (variable in number) in the video. (4) **FLAT w/o comment**: We implement the Shankar et al. (2019) approach which takes as input, a sequence of video frames and discovers a single level of concepts without any hierarchy. (5) **FLAT w/ comment**: This is a modification of the above baseline where we also decode comments from the single-level concept discovered and use that to give the model an extra signal which will help it in discovering better aligned concepts. See Figure 6(C) (in Appendix) (6) **Clustering**: Given an input sequence of frames, we define the weight function based on their temporal position in the sequence and also the $L_2$ distance between the frame embeddings. Then we use standard K-means algorithm (K=4) to cluster the frames based on the weighting function defined and use the clusters formed to predict the temporal boundaries. (7) **GRU_Seg**: Instead of predicting end time stamps of each segment (as in GRU-supervised), the decoder is trained to predict/assign identical ids to frames which are part of the same segment. Further, the model's decoder is trained to assign different ids to frames part of different segments while frames not part of any meaningful segment in the ground truth are trained to have a default null id - 0. During inference, continuous subsequence of frames predicted to be having same id are considered as part of one segment and different predicted segments are extracted accordingly (frames predicted to be having null ids are ignored).

For baselines, we predict only one level of concepts and hence refer to them as high-level concepts. Table 1a summarises and compares the TW-IoU computed between ground truth time stamp annotations and predicted/discovered segments. As can be seen, UNHCLE achieves the best TW-IoU compared to all other baselines. It can be noted that discovering only one (high) level of concepts (FLAT) results in low TW-IoU (for High-Level Concepts). However, UNHCLE discovers concepts that align better with the ground truth events (UNHCLE performs $\sim 23\%$ better) compared to FLAT. Further, the GRU-supervised baseline performs better than FLAT but UNHCLE outperforms ($\sim 15\%$) the GRU-supervised baseline which shows that predicting such concepts is difficult even for a supervised baseline. Even though supervised **GRU_Seg** achieves better TW-IoU of 53.1 compared to high level concepts discovered by UNHCLE w/o comment as well as UNHCLE with comment, UNHCLE with comment (which is unsupervised) performance (TW-IoU 47.4) is still comparable with **GRU_Seg**. However, TW-IoU corresponding to low level concepts discovered by UNHCLE is better than **GRU_Seg**.

---

[3]We use YouCook2 since each recipe comprises of multiple conceptual segments unlike Chess dataset (each game is just an opening + variation) which we instead use for evaluating label prediction ability later.

| Method | TW-IoU | |
|---|---|---|
| Random | 28.2 | |
| EQUALDIV | 31.7 | |
| GRU-supervised | 22.8 | |
| FLAT w/o comment | 14.4 | |
| FLAT w/ comment | 14.8 | |
| Clustering | 32.2 | |
| GRU_Seg | **53.1** | |
| | *high-level* | *low-level* |
| UNHCLE w/o comment | 37.4 | 58.7 |
| UNHCLE w/ comment | **47.4** | **59.7** |

**(a)**

| Frames Used | TW-IoU | |
|---|---|---|
| | *high* | *low* |
| 200 | 37.4 | 58.7 |
| 64 | 33.4 | 34.0 |
| 32 | 13.1 | 12.7 |

**(b)**

**Table 1: (a)** TW-IoU scores for single level concepts predicted by the baselines along with the TW-IoU scores for both the high-level and low-level concepts for predictions from our proposed technique UNHCLE **(b)** Comparison showing the trade-off between number of down-sampled frames and TW-IoU for UNHCLE

### 3.3.3 EFFECT OF GUIDANCE THROUGH COMMENTARY

In this section, we discuss the improvement in the quality and precision of conceptual segments discovered by UNHCLE using video commentary to guide our model training on YouCook2 dataset. To study this, we compare UNHCLE without comment which discovers concept hierarchy using just frame observations in a video against UNHCLE with comment which additionally uses commentary as guide (as in Figure 2). Table 1a compares the TW-IoU of segments obtained corresponding to High Level Concepts for each of these two models. As can be seen, using commentary significantly improves the TW-IoU by $\sim 10\%$ which establishes that using commentary enables UNHCLE to detect precise boundaries for segments corresponding to various concepts. However, we find only a marginal improvement of $\sim 1\%$ in TW-IoU corresponding to low level concepts. Further, it can be seen in Figure 5 that using commentary allows the model to better understand how to merge lower-level concepts appropriately into higher-level concepts - "*heat a pan, add 1 spoon oil and prosciutto to it*" separately without merging it with subsequent segments.

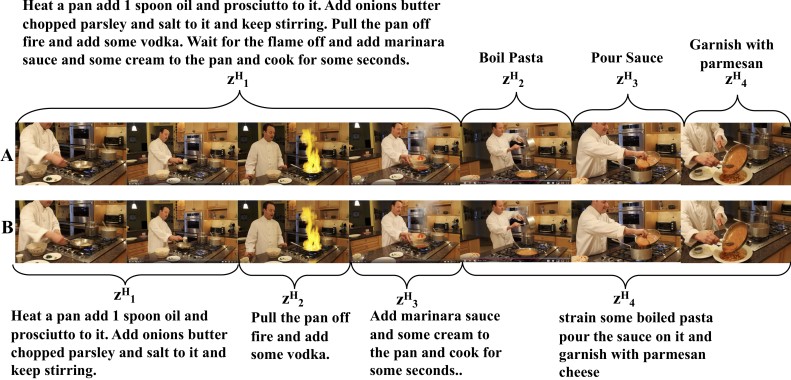

**Figure 5:** In this figure, we show that using commentary as guide during training, UNHCLE learns to better combine low-level concepts to form high-level concepts which are better aligned towards our ground-truth annotations. A refers to the concept segments discovered w/o comments and B refers to the one with comment.

### 3.3.4 LABEL PREDICTION TASK FOR CHESS OPENINGS

We further evaluate UNHCLE on the task of label (name of opening + variation) prediction corresponding to hierarchical concepts discovered. Since recommending Chess Openings comprises of a label for each sequence of moves in an opening, we train UNHCLE w/ commentary (Figure 2). Since there are at max only two segments corresponding to opening move and variation move in the dataset we use **label prediction accuracy** as our metric here instead of *TW-IoU*. There are 300 distinct labels and our model achieves **78.2%** accuracy which shows that it is able to correctly discover the correct

segmentation between openings with different variations as hierarchical concepts and associate them with the correct label in most of the test samples.

### 3.3.5 EFFECT OF SAMPLING RATE ON THE QUALITY OF HIERARCHY

For YouCook2 we down-sample 200 frames from the original 500 frames provided in the dataset and use that as our input to UNHCLE. We analyze the trade-off between time and performance if we further down-sample the frames. This also provides us a better insight into how much granular information regarding the video do we need to be able to discover better hierarchies. Table 1b shows the results obtained for our experiment. Interestingly, we don't observe a linear drop in performance by reducing the number of frames. Even with just 64 frames, the results beat the other baselines.

### 3.4 ABLATION EXPERIMENTS & VISUAL ORDERING TASK

We performed more ablation experiments to show the need for the modules and the losses used in our model. We removed the *soft-DTW*$(z_{1:u}^L, c'^L_{1:u'})$ loss from our UNHCLE (w/o comment) model in Figure 6(A) (in Appendix) to highlight the importance of this loss that guides traversing back and forth across the concept hierarchy. This loss guides the alignment between the initially generated low-level concepts ($z^L$) and the low-level concepts ($c'^L$) extracted back from high-level concepts ($z^H$). Removing this loss reduces the TW-IoU scores drastically thus proving it's necessity .See UNHCLE w/o comment w/o low-align loss in Table 2a.

We also choose a simplified version of the UNHCLE w/o commentary model shown in Fig. 6(A) (in Appendix), where we remove the $OCA^{LOW}$ modules and directly output $z^H$, which looks simpler, but this removes the extra alignment signal as mentioned in the above point. We see this results in the drop of TW-IoU (Table 2a), thus confirming our need for the modules used. We call this the *Direct Hierarchy* baseline for which the system diagram is also provided in Figure 6(B) (in Appendix).

To show the effectiveness of the concepts discovered by UNHCLE we use them to perform a Visual Ordering Task. The task is that given a sequence of frames as input, our trained model (with frozen weights) should discover the high-level concepts associated with those frames and can be used to predict whether or not the given sequence of frames are in correct/meaningful order (binary classification). We use a simple 1-layer GRU to do so. To create the training data for this we take examples from the YouCook2 (Zhou et al. (2017)) dataset and randomly shuffle the sequence of frames creating 10 negative examples for each positive sample in the dataset. We finally evaluate this on our testing data after performing the same steps mentioned above and show our results in Table 2b. We report both the Accuracy and the F1 scores. As we can see that there is a significant gain of **9%** in the F1 score and **2%** in Accuracy using our model UNHCLE over the FLAT baseline.

| Model Variant | TW-IoU | |
|---|---|---|
| | *high* | *low* |
| UNHCLE w/o comment | 37.4 | 58.7 |
| Direct Hierarchy | 20.3 | 28.6 |
| UNHCLE w/o comment w/o low-align loss | 18.9 | 28.7 |

**(a)**

| Model Used | Metrics | |
|---|---|---|
| | *Accuracy* | *F1* |
| FLAT | 88.9 | 2.13 |
| UNHCLE | **90.9** | **14.9** |

**(b)**

**Table 2:** **(a)** TW-IoU scores for ablation experiments for both high-level and low-level concepts prediction **(b)** Comparison between the Shankar et al. (2020) FLAT baseline and our proposed UNHCLE on the designed Visual Ordering Task.

## 4 CONCLUSION

We show that our approach (UNHCLE) discovers concepts and organizes them in a meaningful hierarchy using only demonstration data from chess openings and cooking. We also show that this discovered hierarchy of concepts is useful in predicting textual labels and temporal concept segmentations for the associated demonstrations. It would be interesting to use the concept hierarchy discovered by UNHCLE to generate a curriculum where lower-level concepts would be taught first followed by higher-level concepts. Another good direction would be to extend UNHCLE to learn different types of relationships between concepts without supervision.

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

## A    MORE DETAILS

We provide all the modified system diagrams for all the ablation experiments mentioned in Section 3.4 for clarity.

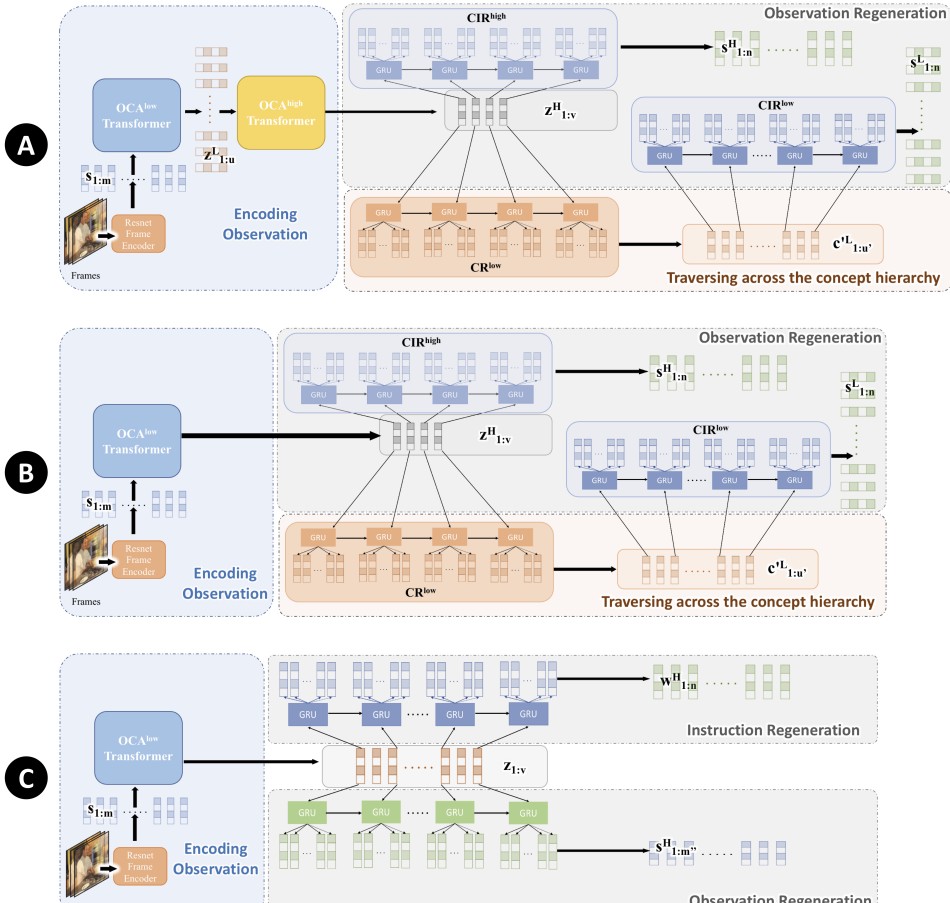

**Figure 6:** Three model variants are shown here. **(A)** refers to UNHCLE w/o comment. **(B)** refers to the Direct Hierarchy variant where we directly predict the high-level skill using OCA transformer. **(C)** refers to FLAT baseline with comment.

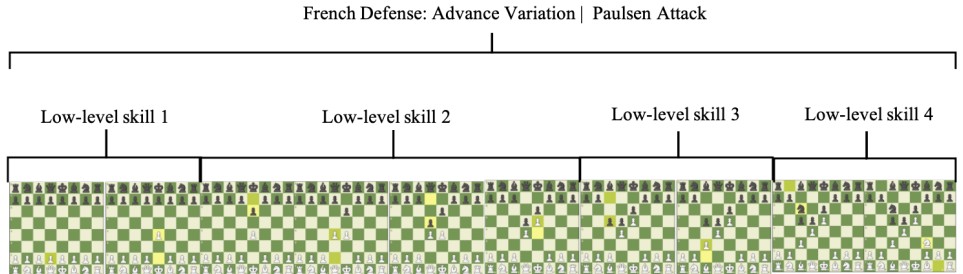

**Figure 7:** Skills discovered in the French defence.

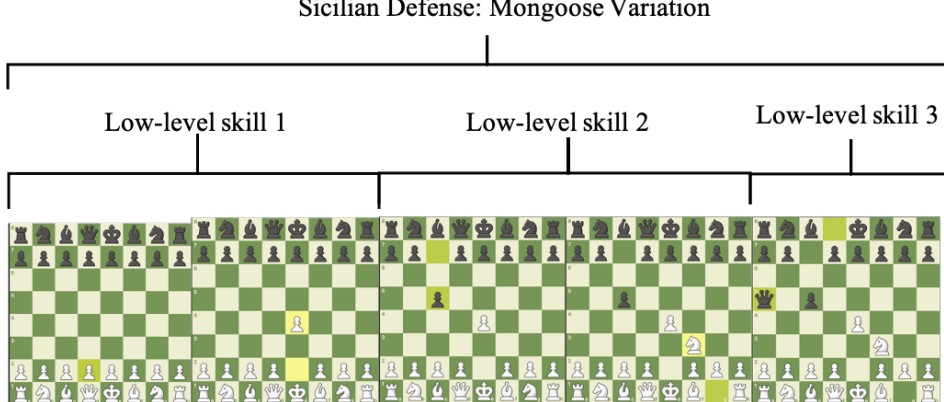

**Figure 8:** Skills discovered in the Sicilian opening.

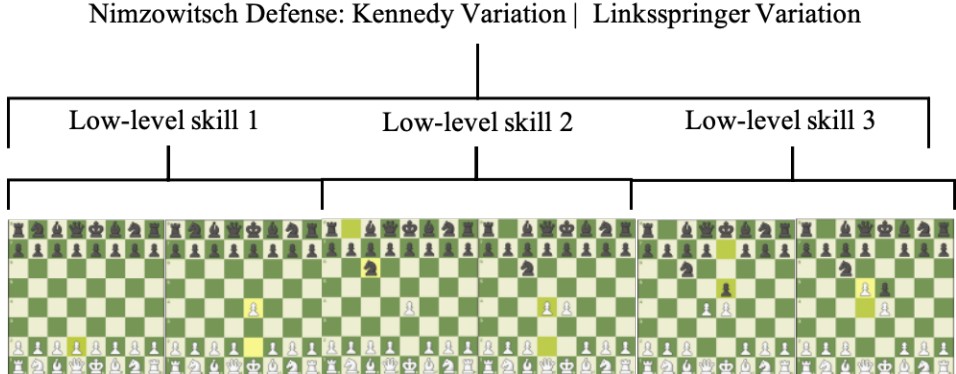

**Figure 9:** Skills discovered in the Nimzowitsch Defense: Kennedy Variation :Linksspringer Variation opening.

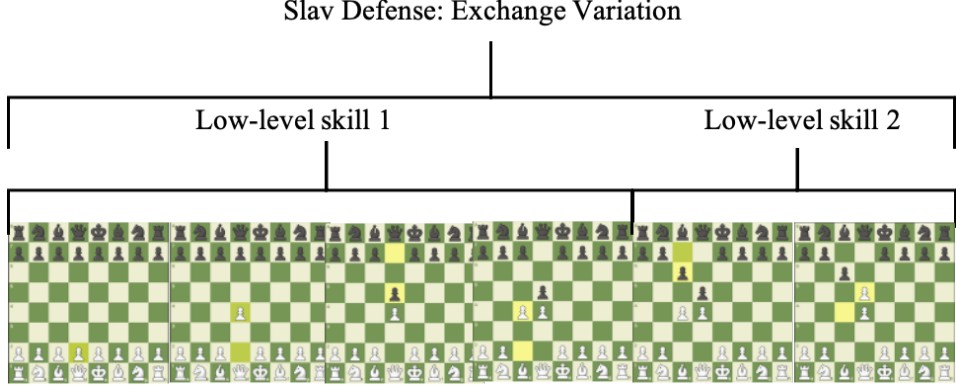

**Figure 10:** Skills discovered in the Slav Defense: Exchange Variation opening.

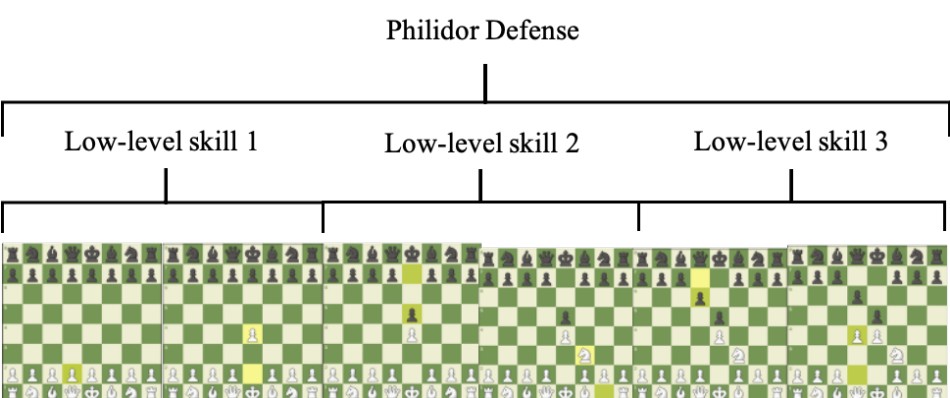

**Figure 11:** Skills discovered in the Philidor Defense opening.

