# OpenReview forum: "Unsupervised Hierarchical Concept Learning"
_ICLR.cc/2021/Conference — Reject_

### Official Review · AnonReviewer1 · 2020-10-22
**Recommend rejection**

**Rating:** 4
**Confidence:** 4

**Review:**

This paper presents a method to unsupervisedly discover structure in unlabeled videos, by finding events at different temporal resolutions.

### Strengths:
- The paper focuses on the important problem of exploiting weakly labeled video data, by exploiting its structure, for example by recovering temporal structure in an autoencoder fashion.
- Use of multiple modalities to cross-supervise each other.
- Code is available.

### Weaknesses:
- The concept of hierarchy is not well defined or well motivated. While most hierarchical papers refer to hierarhies of concepts, the hierarchy considered in this paper is much weaker as a hierarchy, and it refers to subactions within longer actions (not actions that are specific instances of more abstract actions). While this way of understanding hierarchy can be valid, it is never explained or motivated in the paper, or even compared to the standard way of understanding it.
- The overall motivation for the method has a lot of gaps. For example:
	- Regeneration of low level concepts from high level concepts: what are we expecting from a network that moves from a "high in the hierarchy" concept to a "low in the hierarchy" concept? Should we expect the network to randomly select one subconcept? The specific information is not there, the problem is ill-defined (for example, we can go from "cat" to "animal", but not from "animal" to "cat"). How are we expecting any reconstruction? The paper does not provide any justification or intuition.
	- Why are the authors using those specific pairs in the L_dyn term? (last line of page 4 -- I suggest adding equation numbers). Apart from no motivation, there are no ablations showing that those are the correct pairs.
	- Why is the "low" case even necessary? Wouldn't it be possible to train only with the "high" one? This would probably imply rethinking some of the losses, but overall the method would look very similar. This is, the idea of hierarchy would disappear, but this idea is not used in the experiments anyway.
	- What is the motivation for the two modalities? I can understand it can help, but it is not central to the method. This is not necessarily bad, but it requires some explanation.
- The explanation revolves around demonstration data. It is unclear why demonstration data is important for this method, and why it is not general for any human action. The introduction explaining demonstrations in a robotics scenario does not feel related to the content of the paper. For example, a lot of stress is given to "agents" interacting in "environments".
- Some terms introduced in the paper would benefit from a change. For example, a "concept" in the paper is actually an "event", not a "concept". This is more in line with the literature, for example the dataset they use labels that as "event".
- Results on chess are hard to believe. Do the authors think that the system has really learned (unsupervisedly) to identify interesting openings? It could instead be learning strong biases like length of openings in the dataset.
- Quantitative results are not convincing. How does FLAT and even the supervised method perform much worse than the two trivial baselines (random and equal division)? Does FLAT use text data?
- Conceptual assertions that I do not believe to be true: "under a concept, the sequence of frames is nearly deterministic". This is not true, there are nearly infinite ways of having a sequence of frames (video clip) depicting how to crack an egg, for example. Different background, different way of performing the action, different elements in the scene, speed of the action, point of view, etc. This is related to the "regeneration" point above.

### Additional comments and questions:
- Figure 2 is hard to understand. What is it exactly representing? What should we learn from it?
- Does the algorithm have any "motivation" to not predict always a single concept per sequence?
- What is the relationship between u and v?
- Have you tried smaller networks? 8 layers and 8 heads just for the Encoder seem like a very big model for such a relatively simple text.
- In the first paragraph of page 8 the paper mentions that there is only marginal improvement in low level concepts. How are these evaluated? As far as I could understand, there was only GT available for the high level ones.

### Final recommendation
Overall, I believe the paper as it stands is not ready to be presented to ICLR and I recommend a rejection.

---

> ### Author Response · Authors · 2020-11-17
> **Addressing the raised concerns below.**
>
> We thank the reviewer for their insightful feedback.
>
> **The concept of hierarchy:** In our paper, one can think of low-level concepts as small duration events, and high-level concepts are composed of these lower-level concepts and are hence longer duration events. Also, both(low/high-level concepts) repeatedly occur across several demonstrations. We would address this in our paper.
>
> Explanations
> * Moves from a "high in the hierarchy" concept to a "low in the hierarchy"?:
> One can think of these high-level concepts as autoencoding the low-level concepts $z^L$. Therefore to reconstruct the concepts that high-level $z^H$ encoded, we regenerate low-level concepts $z'^{L}$ and calculate its loss with the encoded low-level concepts $z^L$. However, to make sure all the low-level $z^L$ don’t get encoded into one high-level concept $z^H$, we decode a fixed number of $z'^{L}$ from each $z^H$ (4 $z'^{L}$ per $z^H$) using GRU. We then use TW-IOU between the incoming 8 $z^L$ and the 16 decoded $z'^{L}$. This helps make the network encode a variable number of  $z^L$ into each $z^H$.  We do this with the expectation that the network learns to pack the right set of $z^L$ per $z^H$.
> *Should we expect the network to randomly select one subconcept?:
> We don’t think the abstraction into high-level is absolute like cat -> animal. The concepts should contain a context in which it played out, so more like cat->(animal with fur that meows). In our setup, this could mean that the $z^H$ for ”slicing a carrot and boiling it” and “cutting a radish and boiling in a pan” might not be the same but would be close together in the $z^H$ space.
> *Why are the authors using those specific pairs in the L_dyn term?:
> We performed more ablation experiments to show the need for the modules and the losses used in our model. We removed the soft-dtw($c'^{L}, z^L$ ) loss from our UNHCLE (w/o comment) model to highlight the importance of this loss that guides traversing back and forth across the concept hierarchy. This loss guides the alignment between the initially generated low-level concepts $z^L$ and the low-level concepts $c'^{L}$ extracted back from high-level concepts  $z^H$. Removing this loss reduces the TW-IoU scores drastically, thus proving its necessity. See Table 2 (a) (page 9).
> We also choose a simplified version of the UNHCLE w/o commentary model shown in Fig. 6(A) (in Appendix), where we remove the $OCA^{low}$ modules and directly output $z^H$, which looks simpler, but this removes the extra alignment signal as mentioned in the above point. We see this results in the drop of TW-IoU (Table 2 (a) (page 9)), thus confirming our need for the modules used.
> *“Wouldn't it be possible to train only with the "high" one?”:
> Yes, it’s definitely possible, and we’ve already included that in our baselines. Please refer to the FLAT baseline (Shankar et al.), which exactly does the same and tries to discover a single-level concept without hierarchy. But as we can see, it falls behind a lot in terms of TW-IoU score w.r.t to our model UNHCLE (gain~20%).
> *What is the motivation for the two modalities?:
> A video sequence can be organized into a hierarchy in multiple ways. Now validation of the discovered hierarchy can only be done by humans. However, only a few of these discovered hierarchies would be acceptable to humans. Therefore, to learn human-acceptable hierarchies, we use language modality to guide our unsupervised model. This discovers hierarchies that are more aligned towards the human annotations and hence verified easily. Our paper shows the effect of guidance qualitatively through commentary in Section 3.3.3.
> *explanation revolves around demonstration data:
> Shankar et al. ICLR 2020 and few other recent works like Eysenbach et al. (2018) and Sharma et al. (2019), aiming to discover temporal concepts make use of agents interacting with an environment to learn these concepts. This means they have access to large/unlimited amounts of data from the environment. In contrast, demonstration video has only a limited amount of data to learn from. We wanted to highlight this difference. Our model UNHCLE doesn’t require any environment to discover the hierarchical concepts, which is vital for real-world demonstration datasets like YouCook2, which lacks an environment to test on.
> Nothing restricts UNHCLE from working on general human action videos.
> *A "concept" in the paper is actually an "event":
> Yes, these are events in the video but calling them concepts felt more natural for other domains like chess.
> *Do the authors think that the system has really learned to identify interesting openings?:
> Our approach learns to discover chess moves, which are often repeated in succession, and clubs them into single concepts. This is why UNHCLE can identify interesting openings based on similar moves, which are often played together to create an opening. We are adding more chess results to show it is not biased by length. **Added Fig 9,10,11 in appendix**

---

> > ### Author Response · Authors · 2020-11-17
> > **Continuation to address further points discussed by the reviewer**
> >
> >
> > * Quantitative results are not convincing
> > We want to highlight that the baseline FLAT is the current state-of-the-art in unsupervised temporal clustering (Shankar et al. ICLR 2020), for which we have already shown the comparison. The EQUALDIV baseline performs better than FLAT due to the distribution of ground-truth annotations in the YouCook2 data. In Zhou et al., looking at Fig. 3(b) ( Segment Duration Distribution graph), the graph is skewed with low variance in segment duration, which clearly suggests that most segment time durations have similar values. This affects the results we report for EQUALDIV in Table 1 (page 8). in our paper.
> > GRU-supervised not outperforming the simple baseline can be attributed to the fact that we have varying segments in each video in the dataset. We probed the predictions obtained from this baseline and found out that in most of the cases, the supervised baseline predicted fewer timestamp segmentations than present in the ground-truth and consolidated the rest of the video into a single concept. This penalizes the TW-IoU score since, by definition, it will assign higher scores to predictions having a more detailed split among concepts aligned with the ground-truth rather than those which consolidate the entire sequence in a single concept.
> > Another finding is that FLAT (Shankar et al.) performs much worse than other baselines, whereas UNHCLE shows a significant gain in TW-IoU. This suggests that UNHCLE benefits from the **hierarchical way of discovering concepts**. We have also added baselines for both FLAT with and without using text data in Table 1(a) (page 8).
> > * "under a concept, the sequence of frames is nearly deterministic. This is not true":
> > We agree with this comment, and we didn’t use the right words to express our thoughts.  We think that depending on the prior conditions, the sequence of frames under a concept could be of limited variations. Each concept given a context or prior conditions can only play out in a limited number of ways. We will update this in the paper.
> >
> > Additional Comments Response
> > * Figure 2 is hard to understand:
> > In short, our approach takes in a sequence of frames and the entire commentary at once and uses a Transformer to predict low-level concepts. We traverse across the hierarchy and provide these low-level concepts as input to another transformer to predict high-level concepts.  To provide another signal to our unsupervised model to discover better-aligned concepts, we re-convert these high-level concepts back to low-level and take alignment loss between the earlier discovered low-level concepts and the newly discovered ones. (We have added ablations to show this is important. Refer Section 3.4). This can be treated as an auto-encoder mechanism in concept hierarchies. We also reconstruct the input frames vectors and commentary using both these high and low-level concepts and use alignment loss between them, which can be looked at as another auto-encoder type mechanism in an abstract sense.
> > *"motivation" to not predict always a single concept per sequence?:
> > We call each $z^L$ and $z^H$ as a concept, and our model cannot predict a single concept per sequence because the total number of observations is much larger than what each $z^L$ can predict. Each $z'^{L}$ predicts a fixed number (25) of observation-vectors, and each $z^H$ predicts a fixed number(4) of $z'^{L}$. Since the total number of observations is much larger, like 200, so one $z^L$ concept cannot reconstruct the entire sequence. Similarly, one $z^H$ cannot encode the entire $z^L$ sequence
> > *What is the relationship between u and v?:
> > As we mentioned in Section 3.2, we keep the max limit for **u** as 4 and for **v** as 16, which is chosen based on the lengths of the GT annotations available for YouCook2.
> > *Have you tried smaller networks?:
> > We did try out one smaller variant having 4 layers and 4 heads, and we observed a drop in performance of our model caused by that. Adding the ablation here -
> > TW-IoU (high) = 21.9 ; TW-IoU (low) = 28.7 for the UNHCLE w/o comments variant.
> > *only marginal improvement in low-level concepts. How are these evaluated?:
> > Yes, the YouCook2 data doesn’t have GT annotations for low-level concepts, so we evaluate the low-level concepts using GT annotations for high-level concepts itself. Note that this will not harm the TW-IoU score by design. Even with this, we get a marginal improvement in the TW-IoU, which represents the fact that the alignment function chose to align somewhat better low-level concepts with the high-level GT annotations, thus leading to this marginal gain and not that this is the max achievable gain. If we had GT annotations for low-level concepts we would’ve expected the gain to be much higher.

---

> > > ### Author Response · Authors · 2020-11-18
> > > **Added More Chess Results**
> > >
> > > We have added more chess results with Fig 9,10,11 in the appendix. This shows that UNHCLE could find different openings with the same length.

---

> > > > ### Author Response · Authors · 2020-11-25
> > > > **Added Comparison with Stronger Supervised Video Segmentation Baseline**
> > > >
> > > > We design another supervised baseline: **GRU\_Seg** where we modify GRU-supervised baseline to perform video segmentation. Here, instead of predicting end time stamps of each segment (as in GRU-supervised), the decoder of **GRU\_Seg** is trained to sequentially predict/assign identical ids to frames which are part of the same segment. Further, the model’s decoder is trained to assign different ids to frames part of different segments while frames not part of any meaningful segment in the ground truth are trained to have a default null id - 0. During inference, continuous subsequence of frames predicted to be having same id are considered as part of one segment and different predicted segments are extracted accordingly (frames predicted to be having null ids are ignored). **GRU\_Seg** is capable of predicting variable number of segments given a video. The TW-IoU obtained by matching ground truth segments with segments predicted by **GRU\_Seg** is 53.1
> > > >
> > > > It can be seen that even though supervised segmentation baseline **GRU\_Seg** achieves better TW-IoU compared to high level concepts discovered by UNHCLE w/o comment (TW-IoU 37.4) as well as UNHCLE with comment (TW-IoU 47.4), UNHCLE with comment (which is unsupervised) performance (TW-IoU 47.4) is still comparable with **GRU\_Seg** (TW-IoU 53.1). However, UNHCLE is capable of discovering hierarchies within the events in a video as a result of which, TW-IoU corresponding to low level concepts of UNHCLE (TW-IoU 58.7) outperforms **GRU\_Seg**. This discussion (page 7) and results (Table 1(a), page 8) have been added to the paper.
> > > >
> > > > We have also added comparison with unsupervised k-means clustering baseline in Table 1 (a), page 8 (TW-IoU 32.2).

---

### Official Review · AnonReviewer2 · 2020-10-28
**Somewhat limited approach and experiments**

**Rating:** 4
**Confidence:** 4

**Review:**

This paper addresses the problem of extracting a hierarchy of concepts in an unsupervised way from demonstration data. The authors present a Transformer-based concept abstraction architecture called UNHCLE and show how it discovers meaningful hierarchies using datasets from Chess and Cooking domains. In particular, the model is designed to function without specific temporal supervision, which makes it potentially practical for real-world applications.

Pros.
+ problem clear stated with potential practical implications
+ experiments on real-world video datasets

Cons.
- The results are evaluated using a time-warping based metric. I am not convinced this metric provides a faithful assessment of the capability of the model. The exact temporal boundaries are also important in discovering the specific concept and ground them back to the visual observations. TW-IoU will not take this into consideration.
- The novelty of the proposed model is limited. The two-level hierarchy has been seen a lot in relevant fields such as video recognition. So the concept by itself is not new. Introducing a two-level hierarchy into concept learning is also not new. In fact, I find it somewhat arbitrary that the authors decide to set the number of hierarchies to two. The cooking video dataset does not have such annotations so hypothetically the one can set an arbitrary number for the levels of hierarchy. Can the authors explain why this two-level hierarchy is chosen and how it can generalize to other tasks?
- Comparison to works in unsupervised clustering, video segmentation, and change point detection. While the authors form this work as concept learning, the problem nevertheless relates to these research topics. The authors may need to look upon those for a better variety of baselines and also evaluation metrics.
- Experiment results are weak. The proposed model's performance does not exceed the simple equal division baseline a lot when 64 frames are sampled for each video.

---

> ### Author Response · Authors · 2020-11-17
> **Clarification around experiments and hierarchical approach**
>
> * TW-IoU takes into account the original temporal boundaries. For every interval in the ground-truth segment, the time-warped alignment function finds the optimal mapping for that interval to one or multiple intervals in the predicted segments by UNHCLE. This is done for every interval in the ground-truth segment, considering all exact temporal boundaries, and all individual values are added to report the final TW-IoU.
> Also, we would like to point out the benefit of TW-IoU as an evaluation metric. For example: suppose the original segment is [0, a,a+b] where each number represents the ending times of ground-truth concepts (The first concept starts at time 0 ends at time “a”, and the second starts at “a” and ends at “a+b”). Now consider two different sets of aligned predictions : (i) [0,c+d] (ii) [0,c,c+d]. We assume c > a and d > b and all a,b,c,d are positive real numbers without any loss of generality. (Trivial to show the other case). So for (i) TW-IoU = (a+b) / (c+d) and for (ii) a/c + b/d. Now it’s trivial to show that (ii) > (i) under the given assumptions. We would expect this to happen since (ii) is a better prediction as it breaks down the original high-level input to find 2 low-level concepts, whereas (i) doesn’t break it down hierarchically. Thus we would want our proposed metric TW-IoU to perform better in case (ii), which is exactly what will happen. We hope this clears out the motivation behind this. (Also note that the alignment function in TW-IoU works sequentially, i.e., it won’t align any interval with the current ground-truth being processed until all previous intervals have been aligned.)
> * Regarding the two-level hierarchy: We used a different approach to establish a hierarchy in a temporal fashion. This time-warping Soft-DTW has not been used to create a temporal hierarchy before. The first work to use DTW to create single-level skills/concepts was Shankar et al. ICLR 2020. We demonstrate that a temporal hierarchy can be created in an unsupervised manner. A variable number of lower concepts ($z^L$) can be clubbed together into the next higher-level concept ($z^H$), subsequently decoding $z^H$ back to $z'^L$.  E.g., we create four $z^H$ from eight $z^L$, and then each $z^H$ is decoded to a limited (4 in the paper) number of $z'^L$, creating a total of 16 $z'^L$. We then use TW-IOU in between the incoming 8 $z^L$ and the 16 decoded $z'^L$. This helps make the network encode a variable number of  $z^L$ into each $z^H$. This is further validated through the ablation “Direct Hierarchy” shown in Table 2 (a) (page 9). The corresponding architecture is shown in Fig 6 (B)
> This method of creating a high-level temporal concept is not limited to two-levels but can be easily scaled to any hierarchy level by repeating the encoding process mentioned above. To compare FLAT with high-level concepts obtained when done hierarchically, 2 is a natural choice for hierarchy. Further, we had to limit it to two-level because of a lack of GPU memory to fit our network activations.
> * Comparisons: We have added comparisons with unsupervised clustering (Table 1 (a) - page 8). We are in the process of adding comparisons with more baselines.
> * “Model's performance does not exceed the simple equal division by a lot”: The equal division baseline was created for 200 frames video and cannot be compared with UNHCLE using 64 frame video. The 64 frame video has too little information for UNHCLE to work with. Further, EQUALDIV gives reasonable results due to the nature of ground-truth annotations available in the YouCook2 data. In Zhou et al., looking at Fig. 3(b) (Segment Duration Distribution graph), the graph is skewed with low variance in segment duration, which clearly suggests that most segment time durations have similar values.

---

> > ### Author Response · Authors · 2020-11-25
> > **Added Comparison with Stronger Supervised Video Segmentation Baseline**
> >
> > We design another supervised baseline: **GRU\_Seg** where we modify GRU-supervised baseline to perform video segmentation. Here, instead of predicting end time stamps of each segment (as in GRU-supervised), the decoder of **GRU\_Seg** is trained to sequentially predict/assign identical ids to frames which are part of the same segment. Further, the model’s decoder is trained to assign different ids to frames part of different segments while frames not part of any meaningful segment in the ground truth are trained to have a default null id - 0. During inference, continuous subsequence of frames predicted to be having same id are considered as part of one segment and different predicted segments are extracted accordingly (frames predicted to be having null ids are ignored). **GRU\_Seg** is capable of predicting variable number of segments given a video. The TW-IoU obtained by matching ground truth segments with segments predicted by **GRU\_Seg** is 53.1
> >
> > It can be seen that even though supervised segmentation baseline **GRU\_Seg** achieves better TW-IoU compared to high level concepts discovered by UNHCLE w/o comment (TW-IoU 37.4) as well as UNHCLE with comment (TW-IoU 47.4), UNHCLE with comment (which is unsupervised) performance (TW-IoU 47.4) is still comparable with **GRU\_Seg** (TW-IoU 53.1). However, UNHCLE is capable of discovering hierarchies within the events in a video as a result of which, TW-IoU corresponding to low level concepts of UNHCLE (TW-IoU 58.7) outperforms **GRU\_Seg**. This discussion (page 7) and results (Table 1(a), page 8) have been added to the paper.
> >
> > Further, we would like to highlight that GRU-supervised baseline (TW-IoU 22.8) which we have compared in the paper is essentially change point detection since we train the decoder to sequentially predict the end time stamps of relevant segments in the video and these end time stamps essentially are the change points in the video.

---

### Official Review · AnonReviewer4 · 2020-10-28
**Unsupervised encoder-decoder network in a relatively new topic**

**Rating:** 6
**Confidence:** 3

**Review:**

This paper addresses a relatively new topic to learn the hierarchical concepts in videos and commentary in an unsupervised manner. The authors proposed a hierarchical and transversing encoder-decoder network architecture to tackle the problem, where the pre-trained ResNet32 and BERT are used as feature extractors, transformers and GRUs serve as the concept encoders and decoders. As the network is unsupervised, the starting time and ending time of each concept is quite crucial and the problem was tackled by training using a soft-DTW loss. A metric (time-wrapped IoU) is also proposed for quantitative evaluation. The experiments indicate the effectiveness of the network and the network was tested in different datasets and scenarios.

I rate the paper as weakly accept and here are some questions.
1. The lower-level concepts and upper-level concepts in both visual and language are a little vague without clear definitions. How do you define the level of concepts and how did you balance the levels in video and language? For example in Figure 5, the results without commentary (A) look more reasonable for z^H_1 to contain all the steps for a higher-level concept (say, pan heating) but the model with commentary (B) split the concepts mainly according to the language descriptions. This may also influence the evaluation scores as different work may have different "flavors" to select concepts and the starting/ending time could be different.
2. Besides, since your network is trained across the video and language, how did you train the network without commentary part?
3. You used two different evaluation metrics in two different datasets, which seems that you tend to focus on different aspects in different datasets. This is a bit of bias. Besides, the comparison with other work in the second dataset (Chess Openings) is not provided.

---

> ### Author Response · Authors · 2020-11-17
> **Thank you for the suggestions; Addressing the raised concerns below**
>
> Thank you for your succinct reviews. We clarify each of the issues raised below:
> 1. In our paper, one can think of low-level concepts as small duration events, and high-level concepts are composed of these lower-level concepts and are hence longer duration events. Also, both(low/high-level concepts) repeatedly occur across several demonstrations. We assume that most natural language instructions associated with such demonstrations describe higher-level skills since describing lower-level concepts will make instructions verbose. However, there are cases which have verbose or low-level instructions along with the video. Our solution tries to make sense of these without human intervention.
>     * Regarding balancing the levels in video and language, we don’t do any balancing and use the same commentary and video frames for both levels and let the network decide on the best alignment. So in case of the presence of only high-level comments, the high-level z’s should predict aligned segments, but the low-level z’s end up predicting multiple low-level segments inside that larger high-level comment. And in case only low-level comments are present, the low-level z’s predict aligned segments, but the high-level z’s predict segments that encapsulate multiple low-level comment segments.
>     * Regarding Figure 5, On close inspection of the w/o comment A) we can see that $z^H_1$ encapsulates ‘pan heating’ with ‘adding vodka’ and ‘marinara sauce,’ but this encapsulation might not be reusable/common across multiple recipes whereas in B) It puts them in different segments (identifies pan heating with frying onions separately which is quite common). Therefore B) aligns better with human understanding in terms of both hierarchy and reusability.
>     * "different work may have different "flavors" to select concepts"
> The reviewer has highlighted an important point and is correct in saying that this could change according to the dataset being used. We agree with this because it is subjective and ambiguous at times and would differ according to the datasets. But the motivation of this work is to generate a human-understandable hierarchy, and therefore, such ambiguity would come in. One possible solution is to have a generative model with a new evaluation metric to address the ambiguity, but that is an entirely new work altogether and could be the future direction of our work.
> 2. Training without commentary: The architecture was slightly modified. We have added the figures for the ablation architectures in the appendix in Fig. 6(A).
> 3. We did not evaluate the performance using TW-IoU on Chess Openings dataset since each opening mainly comprises only 2 concepts - opening moves followed by the variation moves. Also, there are no ground truth annotations for segments comprising the opening moves and variation in each dataset sample. Instead, we believed that generating the labels (opening label + variation label) is a better way to measure the model’s capability to identify different openings. We evaluated the same quantitatively through label prediction (in section 3.3.4), which considers both the opening and the variation. As can be seen, our model can generate these labels with decent accuracy (~78%).
>
> Please note that we have also updated the paper with one more baseline, which is FLAT w/ comments, and another experiment on using the discovered concepts using both the FLAT baseline and UNHCLE one to perform a visual ordering task.

---

### Official Review · AnonReviewer3 · 2020-10-29
**The paper provides some insights of an important task, but some details can be further improved.**

**Rating:** 5
**Confidence:** 3

**Review:**

The paper introduces the solution of an important task: hierarchical concept learning(or temporal abstractions) from demonstration data. Specifically, this paper considers 1) unsupervised setting; 2) the hierarchy of concepts, and conducts experiments in two datasets. However, there are some points in the experiment section to be discussed.

The pros of this paper include:
- The task setting of this paper is important and applicable. Especially jointly learning concepts from frames and instructions while considering the hierarchy.
- The proposed model utilizes multiple regeneration module for the concept learning, which may also help for other hierarchical-aware tasks.

The cons include (In my opinion, the main weakness is the experiments):
- Some design of model (e.g., Traversing across the concept hierarchy, Observation, and Instruction Regeneration, etc) are not fully estimated in this section: are they really useful? which cases did they corrected? The experiments make the previous model design somewhat difficult to evaluate.
- The selected baselines are also weak. Even the **Random**(Random seg) and **EQUALDIV**(Equal Divided) perform much better than **GRU-supervised**. I believe more powerful baselines should be proposed, otherwise, the contribution of the paper is hard to be recognized.

---

> ### Author Response · Authors · 2020-11-17
> **Detailed Ablation Experiments**
>
> We would like to thank you for your insightful comments. We address each point below. We performed several ablation studies to evaluate each of the components’ necessity, the results of which we were unable to add due to the page limit. We enlist them below and will add them to the appendix of the paper:
>
> 1. Instruction Regeneration Ablation: UNHCLE utilizes the implicit hierarchy that exists in language which guides the representation space towards the ground-truth human events. One of the important baselines that we run is the without-commentary version of UNHCLE. A significantly lower TW-IoU score confirms the influence of the instruction regeneration module. Please see “UNHCLE w/ comment” vs “UNHCLE w/o comment” in Table 1 (a) (page 8).
> 2. Hierarchy Traversal Ablation: The baseline, FLAT (Shankar et al ICLR 2020) is an ablation that studies the effects of using the hierarchy traversal module. The architecture of FLAT is similar to that of UNHCLE w/o comment. The approaches differ in only that the latent space learned by UNHCLE is a 2-level hierarchy, whereas FLAT has a single level concept without any hierarchy. A significantly lower TW-IoU score confirms the influence of the hierarchy. Please see “UNHCLE w/o comment” vs. “FLAT” in Table 1 (a) (page 8).
> 3. We performed more ablation experiments to show the need for the modules and the losses used in our model. We removed the soft-dtw($c’^{L}$, $z^{L}$) loss from our UNHCLE (w/o comment) model to highlight the importance of this loss that guides traversing back and forth across the concept hierarchy. This loss drives the alignment between the initially generated low-level concepts ($z^L$) and the low-level concepts ($c’^{L}$) extracted back from high-level concepts ($z^{H}$). Removing this loss reduces the TW-IoU scores drastically, thus proving it’s necessity. See Table 2 (a) (page 9).
> 4. We also choose a simplified version of the UNHCLE w/o commentary model shown in Fig. 6(A) (in Appendix, page 12), where we remove the $OCA^{low}$ modules and directly output $z^H$, which looks simpler, but this removes the extra alignment signal as mentioned in the above point. We see this results in the drop of TW-IoU (Table 2 (a) (page 9)), thus confirming our need for the modules used.
>
>
>    Model Variant   |||||||||||   TW-IoU (high)  |||||   TW-IoU (low)
>
> Without 2 loss terms (3.) ||||||||  18.9  |||||||||||        28.7
>
> Direct hierarchy (4.) ||||||||||||   20.3 |||||||||||        28.6
>
> UNHCLE w/o commentary ||||||    37.4 |||||||||||        58.7
>
>
> 5. We want to highlight that the baseline FLAT is the current state-of-the-art in unsupervised temporal clustering (Shankar et al ICLR 2020) for which we have already shown the comparison. For EQUALDIV baseline, it performs better than FLAT due to the nature of ground-truth annotations available in the YouCook2 data. In Zhou et. al, looking at Fig. 3(b) ( Segment Duration Distribution graph), the graph is skewed with low variance in segment duration which clearly suggests that most of the segment time durations have similar or close values. This affects the results we report for EQUALDIV in Table 1 (page  8) in our paper.
> 6. GRU-supervised does not outperform the simple baseline can be attributed to the fact that we have a varying number of segments in each video in the dataset. We probed the predictions obtained from this baseline and found out that in most of the cases, the supervised baseline predicted fewer number of timestamp segmentations than present in the ground-truth and consolidated the rest of the video into a single concept. This penalizes the TW-IoU score since, **by definition**, it will assign higher scores to predictions having a more detailed split among concepts aligned with the ground-truth rather than those which consolidate the entire sequence in a single concept. This is how we wanted our evaluation metric to behave and hence proposed the same.
> 7. Another key finding in our paper is that FLAT (Shankar et al) performs much worse than other baselines whereas UNHCLE shows a significant gain in terms of TW-IoU. This suggests that UNHCLE benefits from the **hierarchical way of discovering concepts**. This is intuitive from a human perspective, since when we watch something for example - “boiling an egg”. We create a hierarchical arrangement in our mind that “boiling an egg” is associated with “heating water”, “putting egg”, etc. which helps us better understand the task we are performing.
>
> Please note that we have also updated the paper with one more baseline which is FLAT w/ comments and another experiment on using the discovered concepts using both the FLAT baseline and UNHCLE one to perform a visual ordering task.

---

> > ### Author Response · Authors · 2020-11-25
> > **Added Comparison with Stronger Supervised Video Segmentation Baseline**
> >
> > We design another supervised baseline: **GRU\_Seg** where we modify GRU-supervised baseline to perform video segmentation. Here, instead of predicting end time stamps of each segment (as in GRU-supervised), the decoder of **GRU\_Seg** is trained to sequentially predict/assign identical ids to frames which are part of the same segment. Further, the model’s decoder is trained to assign different ids to frames part of different segments while frames not part of any meaningful segment in the ground truth are trained to have a default null id - 0. During inference, continuous subsequence of frames predicted to be having same id are considered as part of one segment and different predicted segments are extracted accordingly (frames predicted to be having null ids are ignored). **GRU\_Seg** is capable of predicting variable number of segments given a video. The TW-IoU obtained by matching ground truth segments with segments predicted by **GRU\_Seg** is 53.1
> >
> > It can be seen that even though supervised segmentation baseline **GRU\_Seg** achieves better TW-IoU compared to high level concepts discovered by UNHCLE w/o comment (TW-IoU 37.4) as well as UNHCLE with comment (TW-IoU 47.4), UNHCLE with comment (which is unsupervised) performance (TW-IoU 47.4) is still comparable with **GRU\_Seg** (TW-IoU 53.1). However, UNHCLE is capable of discovering hierarchies within the events in a video as a result of which, TW-IoU corresponding to low level concepts of UNHCLE (TW-IoU 58.7) outperforms **GRU\_Seg**. This discussion (page 7) and results (Table 1(a), page 8) have been added to the paper.
> >
> > Further, we have added comparisons with unsupervised k-means clustering (Table 1 (a) - page 8) (TW-IoU 32.2)

---

### Decision · Program_Chairs · 2021-01-07
**Final Decision**

**Decision:**

Reject

**Comment:**

Although the paper studies a relevant and important topic, which is about learning of hierarchy of concepts in an unsupervised manner, the reviewers raised several critical concerns. In particular, although the hierarchical structure of concepts is the key idea in this paper, the concept of hierarchy itself is not well explained. How to define the hierarchical level of concepts should be carefully and mathematically discussed. In addition, empirical evaluation is not thorough as reviewers pointed out. Although we acknowledge that the authors addressed concerns by the author response, newly added results are still confusing and more careful treatment is needed before publication. I will therefore reject the paper.

This work reminds me the the topic called "formal concept analysis" (e.g. see [1]), which mathematically defines concepts as closed sets and constructs a hierarchy of concepts in an unsupervised manner. This method can be viewed as co-clustering and also has a close relationship to closed itemset mining. This approach is used in machine learning (e.g. [2]). I think it is beneficial for the authors to refer such existing and well-established approaches to elaborate this work further.

[1] Davey, B.A., Priestley, H.A.: Introduction to Lattices and Order, Cambridge Univ. Press (2002)
[2] Yoneda, et al., Learning Graph Representation via Formal Concept Analysis, 	arXiv:1812.03395